Glyceraldehyde-3-phosphate dehydrogenase Gh_GAPDH9 is associated with drought resistance in Gossypium hirsutum

Geng Shiwei
Li Shengmei
Zhao Jieyin
Gao Wenju
Chen Qin
http://orcid.org/0000-0001-6591-7264 Zheng Kai
Wang Yuxiang
Jiao Yang
Long Yilei
Liu Pengfei
Qu Yanying
Chen Quanjia chqjia@126.com
College of Agriculture, Xinjiang Agriculture University , Urumqi, Xinjiang , China
Abd El-Moneim Diaa
Electronic publication date: 2023 Nov 23
Publication date: 2023
Volume: 11
Electronic Location ID: e16445
Received 2023 Jun 12; Accepted 2023 Oct 21
Copyright: © 2023 Geng et al.
Copyright year: 2023
Copyright holder: Geng et al.
License: This is an open access article distributed under the terms of the Creative Commons Attribution License, which permits unrestricted use, distribution, reproduction and adaptation in any medium and for any purpose provided that it is properly attributed. For attribution, the original author(s), title, publication source (PeerJ) and either DOI or URL of the article must be cited.
License URL: https://creativecommons.org/licenses/by/4.0/

Keywords: G. hirsutum, Glyceraldehyde-3-phosphate dehydrogenase, Drought stress, Phylogenetic analysis, Gh_GAPDH9

Funding: Major Science and Technology Project of Xinjiang Uygur Autonomous Region 2021A02001-3 Xinjiang Agricultural University Crop Science Key Discipline Project XNMH2019002 This research was funded by the Major Science and Technology Project of Xinjiang Uygur Autonomous Region (No. 2021A02001-3) and the Xinjiang Agricultural University Crop Science Key Discipline Project (No. XNMH2019002). The funders had no role in study design, data collection and analysis, decision to publish, or preparation of the manuscript.

==============================
Background

Glyceraldehyde-3-phosphate dehydrogenase (GAPDH) is the central enzyme of glycolysis and plays important regulatory roles in plant growth and development and responses to adverse stress conditions. However, studies on the characteristics and functions of cotton GAPDH family genes are still lacking.

Methods

In this study, genome-wide identification of the cotton GAPDH gene family was performed, and the phylogeny, gene structures, promoter progenitors and expression profiles of upland cotton GAPDH gene family members were explored by bioinformatics analysis to highlight potential functions. The functions of GhGAPDH9 in response to drought stress were initially validated based on RNA-seq, qRT‒PCR, VIGS techniques and overexpression laying a foundation for further studies on the functions of GAPDH genes.

Results

This study is the first systematic analysis of the cotton GAPDH gene family, which contains a total of 84 GAPDH genes, among which upland cotton contains 27 members. Quantitative, phylogenetic and covariance analyses of the genes revealed that the GAPDH gene family has been conserved during the evolution of cotton. Promoter analysis revealed that most cis-acting elements were related to MeJA and ABA. Based on the identified promoter cis-acting elements and RNA-seq data, it was hypothesized that Gh_GAPDH9, Gh_GAPDH11, Gh_GAPDH19 and Gh_GAPDH21 are involved in the response of cotton to abiotic stress. The expression levels of the Gh_GAPDH9 gene in two drought-resistant and two drought-sensitive materials were analyzed by qRT‒PCR and found to be high early in the treatment period in the drought-resistant material. The silencing of Gh_GAPDH9 based on virus-induced gene silencing (VIGS) technology resulted in significant leaf wilting or whole-plant dieback in silenced plants after drought stress compared to the control. The content of—malondialdehyde (MDA) in cotton leaves was significantly increased, and the content of proline (Pro) and chlorophyll (Chl) was reduced. In addition, the leaf wilting and dryness of transgenic lines under drought stress were lower than those of wild-type Arabidopsis, indicating that Gh_GAPDH9 is a positive regulator of drought resistance. In conclusion, our results demonstrate that GAPDH genes play an important role in the response of cotton to abiotic stresses and provide preliminary validation of the function of the Gh_GAPDH9 gene under drought stress. These findings provide an important theoretical basis for further studies on the function of the Gh_GAPDH9 gene and the molecular mechanism of the drought response in cotton.

Introduction

Cotton is one of the most important cash crops in the world and occupies a very important position in economic development (Khan et al., 2020). With global warming, drought has become a major adverse factor affecting the growth and development, geographical distribution, yield and quality of crops (Lobell & Gourdji, 2012). When cotton is subjected to drought stress, it not only causes inadequate boll development but also hinders the transport of photosynthetic products and mineral nutrients to the growth organs, thereby shortening the cotton reproductive period, decreasing yield composition factors and reducing seed cotton yield, leading to decreases in cotton yield and quality (Sun et al., 2021; Loka, 2012). Therefore, improving the drought resistance of cotton in production has become an important task for cotton breeders. The glyceraldehyde-3-phosphate dehydrogenase (GAPDH) protein is an important catalyst for carbohydrate metabolism and participates in plant stress resistance, therefore, the study of GAPDH gene expression regulation is of great significance for improving the drought resistance of cotton.

GAPDH is a key enzyme involved in glycolysis and is present in almost all organisms (not limited to plants) (Sirover, 2011). This enzyme catalyzes the conversion of glyceraldehyde-3-phosphate (G3P) to 1,3-bisphosphoglyceric acid in the presence of NAD+ and inorganic phosphate (Howard, Lloyd & Raines, 2011). GAPDH is expressed at high levels in almost all tissues under normal or stress conditions and was once considered a “housekeeping gene” for use in gene expression and protein research (Nicholls, Li & Liu, 2012). GAPDH is one of the most important enzymes in cellular energy metabolism, facilitating the first energy-producing response in the glycolytic pathway. It catalyzes the oxidative phosphorylation of glyceraldehyde-3-phosphate to form nicotinamide adenine dinucleotide (NADH). However, recent studies have shown that in addition to its role in glycolysis, nonglycolytic functions of plant GAPDH related to transcriptional regulation, signal transduction, the cytoskeleton and DNA repair have also been demonstrated, particularly in abiotic stress and growth and development pathways (Hadj et al., 2021; Srivastava et al., 2021; Zhang et al., 2020; Zhang, Zhao & Zhou, 2017; Zhang, Zhang & Yang, 2020; Garcin, 2018; Munoz et al., 2011; Kosova, Khodyreva & Lavrik, 2017). For example, the overexpression of OsGAPC3 in rice resulted in increased tolerance of transgenic lines after salt stress, OsGAPC3 can reduce salt toxicity by regulating hydrogen peroxide (H2O2) levels (Zhang et al., 2011). In aspen, GAPDH gene expression was upregulated after drought stress (Pelah et al., 1997). In wheat, TaGAPDH12 expression was significantly upregulated in shoots after exposure to four abiotic stresses (cold, heat, salt and drought) (Zeng et al., 2016). In Arabidopsis, GAPC1 and GAPC2 interact with plasma membrane-bound phospholipase D (PLD) to mediate the plant response to ABA and water deficits (Guo et al., 2012). In soybean, GmGAPDH14 transgenic plants overexpressing GmGAPDH14 were found to grow better than the control after salt stress, GmGAPDH14 gene-overexpressing transgenic plants had higher superoxide dismutase activity, and the malondialdehyde (MDA) content was lower than that of salt-stressed control plants (Zhao et al., 2023). In addition, GAPDH has been associated with root and pollen development in Arabidopsis (Munoz et al., 2009; Munoz et al., 2010). In summary, the members of the GAPDH gene family have been demonstrated to be associated with drought and salt stress, and their expression levels vary in different tissues or developmental stages.

The continued refinement and publication of plant genomes has facilitated the identification of plant gene families, and analysis of the GAPDH gene family is the key to elucidating its evolution and differentiation, and can clarify its grouping in plant species. To date, eight, 22, 16, 16 and 16 GAPDH genes have been identified in Arabidopsis, wheat, soybean, poplar and kale, respectively (Zeng et al., 2016; Zhao et al., 2023; Vescovi et al., 2013; Hui et al., 2022; Xie et al., 2021). In plants, GAPDH can be divided into phosphorylated and nonphosphorylated types based on biochemical properties (Zeng et al., 2016). Phosphorylated GAPDH isoforms are divided into three classes with different subcellular localizations: GAPA and GAPB (chloroplast), GAPC (cytosol) and GAPCp (plastid) (Marri et al., 2005; Anoman et al., 2015; Mirko et al., 2013). Plant cytoplasm contains nonphosphorylated NADP-dependent GAPDH (GAPN), which belongs to the aldehyde dehydrogenase superfamily and has no close functional and structural relationship with phosphorylated GAPDH, catalyzing the oxidation of Ga3P to 3-phosphoglycerate (3PGA) (Mirko et al., 2013). The N-terminal structural domain (PF00044) and the C-terminal catalytic structural domain (PF02800) of GAPDH have been shown to be essential (Zeng et al., 2016; Zhao et al., 2023; Xie et al., 2021; Miao et al., 2019).

The GAPDH gene family has shown potential importance for involvement in plant stress tolerance, and the evolution, function and classification of this gene family in cotton have not been systematically studied. A comprehensive understanding and analysis of the GAPDH gene family in cotton is necessary, especially to understand potential functions in stress adaptation. In this study, members of the GAPDH family were identified based on the evolutionary relationship of cotton combined with genomic data, and bioinformatics analysis was performed. The chromosomal distribution, evolutionary relationships and cis-acting elements of the GAPDH genes in terrestrial cotton were analyzed. qRT‒PCR was used to analyze the expression of candidate genes in different tissues and under drought treatments in upland cotton, revealing their possible biological functions. The functions of GhGAPDH9 in response to drought stress were initially validated based on RNA-seq, qRT‒PCR, VIGS techniques and overexpression, laying a foundation for further studies on the functions of GAPDH genes. These findings provide an important theoretical basis for further studies on the function of the Gh_GAPDH9 gene and the molecular mechanism of the drought response in cotton. Such studies will further broaden our understanding of the function of the GAPDH genes in plants, provide a basis for further research on the function of these genes in cotton drought resistance, and lay a foundation for subsequent analysis of their function.

Materials and Methods

Plant material

Based on previous research, drought-resistant (Acala1517-08 (which originated as a single-plant selection derived from a cross between B7636, an unreleased breeding line, and LA 887, which is moderately resistant to root-knot nematodes) and Tashigan7 hao (which was bred in the former Soviet Union and preserved by the Xinjiang Bazhou Institute of Agricultural Sciences, numbered ZM-06734)) and drought-sensitive (Xinluzhong36 (which was selected and bred by the Bayingolin Autonomous Prefecture Agricultural Science Research and Xinjiang registration number is 20100108) and Xinluzao26 (which is a new variety cultivated by Xinjiang Tianhe Seed Industry Co., Ltd. in 2006, derived from the variant of Xinlu Zao No. 8)) upland cotton materials were selected, all of which were conserved and provided by the Germplasm Innovation Laboratory of Xinjiang Agricultural University (Fan et al., 2021). The cotton seeds of the four materials were sterilized by soaking in hydrogen peroxide for 4 h, washed in sterile water and soaked for 24 h to germinate the seeds. Then, germination boxes that were 18 cm long, 12 cm wide and 6 cm high were selected, and 50 seeds were placed in each box. The boxes were then placed in a constant-temperature (25 °C) incubator with a light intensity of 1200 lx or more and a relative humidity of 60–80%, set to 8 h of darkness and 16 h of light, and incubation was continued until the cotyledons expanded. The cotton plants were then transferred to white plastic pots that were 60 cm long, 45 cm wide and 20 cm high and cultivated in 50% Hoagland’s nutrient solution. The nutrient solution was changed once every 2 d (to ensure that the nutrient solution did not pass by the roots), three true leaves were allowed to grow, and processing was started. Before treatment, the water in each hydroponic pot was replaced with clean, fresh water, in which incubation was continued for 2 d. Thereafter, the water in each hydroponic pot was replaced with a 15% solution of PEG6000 configured to simulate drought treatment, and root, stem and leaf tissues were collected after 0, 0.5, 1, 3, 6, 12 and 24 h of treatment and stored frozen in a −80 °C refrigerator. Three biological replicates of each treatment were performed. Columbia wild-type Arabidopsis thaliana (WT) was used as receptor material for transfecting the GAPDH9 gene.

Identification and bioinformatics analysis of the GAPDH gene family in cotton

The genomic and proteomic data for G. arboreum (CRI, version 1.0), G. raimondii (JGI, version 2.0), G. hirsutum (HUA, version 1.0) and G. barbadense (ZJU, version 1.1) were obtained from the COTTONGEN (http://www.cottongen.org/) database (Yu et al., 2014). The Cryptomarkov model of the structural domain of the GAPDH genes (PF00044 and PF02800) and HMMER software (http://hmmer.org/) were used for analysis (Potter et al., 2018). Using known GAPDH homologous sequences, machine learning in the algorithm generates a hidden Markov model, which we then use to predict homologous gene sequences in cotton. Such a hidden Markov model is actually a file on the Pfam database with the suffix “hmm” (PF00044 and PF02800) (Potter et al., 2018). After the removal of redundancy, all candidate genes were subjected to structural domain validation in the NCBI-CDD (https://www.ncbi.nlm.nih.gov/cdd/) database (Lu et al., 2020). The number of amino acid residues, relative molecular masses, and theoretical isoelectric points of the upland cotton GAPDH proteins were calculated using ExPASy (http://cn.ExPASy.org/) (Mariethoz et al., 2018) and Plant-mPLoc online software (http://www.csbio.sjtu.edu.cn/bioinf/plant-multi/) to predict the subcellular localization of upland cotton GAPDH proteins (Chou & Shen, 2010).

Phylogenetic and collinear analysis of the GAPDH gene family in cotton

To further understand the evolution and differentiation of the GAPDH genes in terrestrial cotton, the GAPDH protein sequences of different cotton species were compared using Clustal W in MEGA 7 software for multiple sequence alignment, and then a phylogenetic tree was constructed using the proximity method based on the results of the alignment, where the bootstrap value was set to 1,000 (Kumar, Stecher & Tamura, 2016). The resulting phylogenetic trees were embellished using the online tool Evolview (https://evolgenius.info/). Local BLAST searches were performed using GAPDH protein sequences from the four cotton species with the e value set to 1e-10. The results of the comparisons were filtered and analyzed using MCScanX software (Wang et al., 2012b), and the results were plotted using CIRCOS software (Zhang, Meltzer & Davis, 2013).

Chromosomal localization, gene structure and motif analysis of the GAPDH gene family in upland cotton

To further understand the composition of the terrestrial cotton GAPDH genes and their locations on the chromosomes, information on the chromosomal locations of upland cotton GAPDH gene family members was extracted using TBtools software version 1.120 (Chen et al., 2020), and the chromosomal locations of upland cotton GAPDH genes were mapped using MapChart software version 2.32 (Voorrips, 2002). An evolutionary tree of upland cotton GAPDH was constructed by MEGA 7 software to obtain NWK files (Kumar, Stecher & Tamura, 2016). Motif analysis was performed with the MEME program (setting the number of functional domains to 10, http://meme-suite.org/) (Bailey et al., 2009). All of the above results were visualized using TBtools software version 1.120 (Chen et al., 2020).

Analysis of the cis-acting elements in upland cotton GAPDH gene promoters

Transcription factors control transcription by identifying specific DNA sequences in gene promoter regions to guide genome expression and thus regulate plant function. It is generally believed that the 2,000 bp region upstream of a gene is the promoter region of that gene. The 2,000 bp DNA sequences upstream of the GAPDH genes were extracted as promoter sequences based on the positional information of the upland cotton GAPDH genes using the PlantCARE database (http://bioinformatics.psb.ugent.be/webtools/plantcare/html/) to predict possible cis-acting elements, which were visualized using TBtools software (Chen et al., 2020).

RNA-seq analysis

Transcriptomic data from different tissues (calycle, leaf, petal, pistil, root, stamen, stem and torus) of upland cotton and data obtained after drought, cold, heat and salt stress treatments were downloaded from the NCBI SAR (Sequence Read Archive) database (genome sequencing project accession: PRJNA532694). The data were filtered and subjected to quality control using fastp software, and the resulting clean data were used for subsequent analysis. The upland cotton (TM-1) genome (https://www.cottongen.org/species/Gossypium_hirsutum/HAU-AD1_genome_v1.0_v1.1) was used as the reference genome for alignment using HISAT2 software, and String Tie was used to compare the postalignment sequences for expression quantification (Hu et al., 2019). Different RNAs may have different lengths, and the longer the length, the more reads there are. When each RNA is divided by its length, the relative expression of different genes in the same sample can be compared. It’s important to note that various samples may have been sequenced at different sequencing depths, and the deeper the sequencing depth, the greater the number of corresponding reads. Normalizing the results by the number of reads per million in their respective libraries provides a robust measure of the relative expression of the same gene in two distinct samples. The FPKM (fragments per kilobase exon per million fragments mapped) value is the number of reads per kilobase mapped to one exon per million reads in the map. The FPKM method was used to assess gene expression. Heatmaps were created using TBtools software (Chen et al., 2020).

qRT–PCR

The qRT‒PCR procedure was performed according to Li et al. (2022). Total RNA was isolated by referring to the Polysaccharide Polyphenol Total RNA Extraction Kit instructions (DP411; Tiangen, Beijing, China). cDNA was synthesized using a FastKing RT Kit (KR116; Tiangen, Beijing, China). Specific primers were designed using the NCBI Primer Tool, and the primer sequences are shown in Table S1. qRT‒PCR was performed using the ABI 7500 platform and PerfectStart Green qPCR SuperMix (AQ601; TransGen Biotech, Beijing, China). The reaction program was a thermal cycling program at 94 °C for 30 s followed by 40 cycles of 95 °C for 5 s, 57 °C for 5 s, and 72 °C for 34 s (three biological replicates). The internal reference gene was GhUBQ7, and the results were calculated using the 2−ΔΔCt method (Zhao et al., 2021).

VIGS and overexpression

VIGS technology produces RNAi directly in contemporary plants, resulting in significant time and cost savings. It can avoid the drawbacks of traditional transgenic systems and can downregulate the expression of deletion-lethal genes. A 409 bp fragment of GhGAPDH9 was amplified from the cDNA of Acala1517-08 at the EcoR and XbaI sites. pTRV2-GhGAPDH9 was successfully constructed and transformed into Agrobacterium tumefaciens GV3101. The TRV vector-carrying Agrobacterium cells were mixed and injected into the cotyledons of 7-d-old seedlings of the drought-tolerant material Acala1517-08, which was then incubated at 22–25 °C under a 16 h/8 h light/dark cycle after 24 h of protection from light. The silencing efficiency was verified by qRT‒PCR in pTRV2-GhGAPDH9 and control pTRV2-injected plants after the appearance of bleaching in leaves injected with pTRV2-CLA. After the confirmation of silencing, watering was stopped until phenotypic differences appeared between silenced and control cotton. Under drought conditions, the changes in malondialdehyde (MDA), proline (Pro) and chlorophyll (Chl) contents were important indicators to judge the drought resistance of plants. The physiological indicators of pTRV2-GhGAPDH9 and control pTRV2, including MDA, Pro and Chl, were subsequently determined according to Fan et al. (2021). Overexpression can cause the gene expression product to exceed normal levels, enabling observation of the biological behavior of the gene and thus elucidating the function of the gene. A 1,224 bp fragment of GhGAPDH9 was amplified from the cDNA of Acala1517-08 at the SacI sites, and pCAMBIA3301-GhGAPDH9 was successfully constructed and transformed into Agrobacterium tumefaciens GV3101 (Yu et al., 2020). Wild-type Arabidopsis and GhGAPDH9-overexpressing Arabidopsis plants were grown and transplanted, and overexpression of the GhGAPDH9 gene in pure strains was determined using qRT‒PCR. The plants were subjected to natural drought treatment (from the time of treatment, watering was stopped, simulating natural drought conditions), and the phenotypes were observed periodically after the drought treatment and photographed and recorded according to the degree of leaf wilt in Arabidopsis. All primers used in this experiment are listed in Table S1.

Statistical analysis

Excel 2019 was used to collate physiological indicators and qRT‒PCR data, SPSS 26.0 software (SPSS Inc., Chicago, IL, USA) was used to analyze the data by ANOVA, and R language was used for graphing (R Core Team, 2020).

Results

Identification of the GAPDH gene family in cotton

Based on the protein functional domains PF00044 and PF02800, a total of 84 GAPDH proteins encoded by four cotton genomes (G. arboreum, G. raimondii, G. hirsutum and G. barbadense) were identified, and the results of the search were validated in the NCBI-CDD database. The copy number changes in the GAPDH gene family during the evolution of cotton were systematically investigated. The G. arboreum, G. raimondii, G. hirsutum and G. barbadense genomes encoded 14, 14, 27 and 29 GAPDH proteins, respectively, suggesting that there has been no gene loss or chromosomal rearrangement in the GAPDH gene family as a result of chromosome doubling or the evolution of cotton. We named the 27 upland cotton GAPDH1 to Gh_GAPDH27 proteins according to the order of their chromosomal positions. The open reading frames (ORFs) of the upland cotton GAPDH family genes are 858–1,368 bp in length and encode proteins containing 285–455 amino acid residues, which is less variable than the ORF sequence. The relative molecular masses ranged from 31.13 to 48.65 kDa, and the theoretical isoelectric points ranged from 6.32 to 10.18, indicating that the physicochemical properties of the GAPDH family genes did not differ significantly. Subcellular localization analysis of the proteins showed that eight localized to the chloroplasts, 14 to the cytoplasm and five to the cytoplasm/mitochondria (Table 1).

Table 1 Information on the GAPDH gene family in G. hirsutum.

Gene id	Gene name	Open
reading frame/bp	Protein length/aa	Relative
molecular weight(r)/kDa	Theoretical isoelectric point (pI)	Subcellular
localization	
Gh_GAPDH1	Ghir_A01G008950	1,281	426	45.31	9.22	Cytoplasm/mitochondrion	
Gh_GAPDH2	Ghir_A01G009430	1,368	455	48.65	9.35	Chloroplast	
Gh_GAPDH3	Ghir_A03G014410	900	299	32.50	8.22	Cytoplasm	
Gh_GAPDH4	Ghir_A03G014420	858	285	31.13	10.08	Cytoplasm	
Gh_GAPDH5	Ghir_A03G014430	1,002	333	36.62	7.69	Cytoplasm	
Gh_GAPDH6	Ghir_A05G018220	1,368	455	48.43	7.21	Chloroplast	
Gh_GAPDH7	Ghir_A05G035300	1,011	336	36.56	8.19	Cytoplasm	
Gh_GAPDH8	Ghir_A07G024350	1,011	336	36.48	7.62	Cytoplasm	
Gh_GAPDH9	Ghir_A07G025060	1,212	403	43.15	8.68	Chloroplast	
Gh_GAPDH10	Ghir_A09G003700	1,029	342	37.36	6.8	Cytoplasm	
Gh_GAPDH11	Ghir_A10G006690	1,227	408	43.37	8.45	Chloroplast	
Gh_GAPDH12	Ghir_A11G004090	1,023	340	36.95	7.69	Cytoplasm	
Gh_GAPDH13	Ghir_A11G007150	1,014	337	36.65	8.59	Cytoplasm	
Gh_GAPDH14	Ghir_A11G031920	1,287	428	45.63	8.79	Cytoplasm/mitochondrion	
Gh_GAPDH15	Ghir_A13G013130	1,272	423	45.16	10.13	Cytoplasm/mitochondrion	
Gh_GAPDH16	Ghir_D01G009850	1,365	454	48.45	7.45	Chloroplast	
Gh_GAPDH17	Ghir_D02G015800	1,011	336	36.85	7.21	Cytoplasm	
Gh_GAPDH18	Ghir_D04G008460	1,011	336	36.56	8.19	Cytoplasm	
Gh_GAPDH19	Ghir_D05G018230	1,368	455	48.45	7.74	Chloroplast	
Gh_GAPDH20	Ghir_D07G024400	1,011	336	36.46	7.63	Cytoplasm	
Gh_GAPDH21	Ghir_D07G025130	1,212	403	43.16	8.68	Chloroplast	
Gh_GAPDH22	Ghir_D09G003620	1,011	336	36.56	7.72	Cytoplasm	
Gh_GAPDH23	Ghir_D10G008680	1,227	408	43.36	8.45	Chloroplast	
Gh_GAPDH24	Ghir_D11G004030	1,083	360	39.73	6.32	Cytoplasm	
Gh_GAPDH25	Ghir_D11G007100	1,014	337	36.64	8.59	Cytoplasm	
Gh_GAPDH26	Ghir_D11G032310	1,251	416	44.33	9.52	Cytoplasm/mitochondrion	
Gh_GAPDH27	Ghir_D13G013840	1,272	423	45.07	10.18	Cytoplasm/mitochondrion	

To investigate the genomic distribution of the upland cotton GAPDH genes on chromosomes, we studied the chromosomal locations of the Gh_GAPDH family members. The results showed that the 27 Gh_GAPDH genes were distributed on 17 upland cotton chromosomes (Fig. 1), with subgroup A containing 15 GAPDH genes and subgroup D containing 12 GAPDH genes. Previous studies have suggested that G. arboreum and G. raimondii are donor species of the G. hirsutum A and D subgenomes, respectively, although the number of GAPDH genes in the A subgenome of upland cotton is one greater than the number of GAPDH genes in G. arboreum, and the number of GAPDH genes in the D subgenome is two less than the number of GAPDH genes in G. raimondii (Hu et al., 2019). This difference suggests that the GAPDH genes may have been relatively conserved during the evolution of upland cotton and that no tandem duplication or segmental duplication events have occurred in this species, which further suggests that the loss of a particular GAPDH gene may have occurred during evolution of the upland cotton GAPDH genes.

Figure 1 Chromosomal localization of the GAPDH gene in G. hirsutum.

Evolutionary analysis of the GAPDH genes in cotton

To further understand the evolutionary relationships of the upland cotton GAPDH gene family, we constructed an evolutionary tree using the protein sequences of the GAPDH genes from G. arboreum, G. raimondii, G. hirsutum and G. barbadense. In addition, 84 cotton GAPDH protein sequences were classified into three groups, GAPA/B, GAPC and GAPCp, according to isoform classification (Marri et al., 2005; Anoman et al., 2015; Mirko et al., 2013), which were located in Group 1, Group 3 and Group 2, respectively. Upland cotton contained eight GAPA/B genes, 14 GAPC genes and five GAPCp genes (Fig. 2). The number of GAPDH genes in each subgroup was essentially twice as high in G. hirsutum and G. barbadense as in G. arboreum and G. raimondii. This result was consistent with those of the previous analysis and in line with the evolutionary relationships of cotton, which suggests that GAPDH family genes have remained relatively conserved during the evolution of cotton. Although there were relatively few members in Group 2, they have existed throughout the evolution of cotton, suggesting that they may have played an important role in the development of cotton and the response to adverse stress conditions.

Figure 2 Evolutionary tree of the GAPDH gene family in cotton.

The pentagram represents the GAPDH gene of G. arboreum, The triangle represents the G. raimondii GAPDH gene, The round represents the G. barbadense GAPDH gene, The square represents the G. hirsutum GAPDH gene.

Based on gene numbers, chromosome positions and phylogenetic tree analysis, GAPDH was found to be conserved during the evolution of cotton. To investigate the evolutionary relationships of the cotton GAPDH family in depth, we selected upland cotton as the core species and constructed a covariance matrix of G. hirsutum with G. arboreum, G. raimondii and G. barbadense using MCScanX software (Fig. 3). G. hirsutum shared 135 collinear gene pairs with G. arboreum, 123 collinear gene pairs with G. raimondii, and 271 collinear gene pairs with G. barbadense, and G. hirsutum itself exhibited 230 collinear gene pairs. These results suggest that upland cotton GAPDH has remained relatively conserved during the evolution of cotton.

Figure 3 Covariance of GAPDH genes of G. hirsutum with those of G. arboreum, G. raimondii and G. barbadense.

The blue line indicates the collinearity of GAPDH genes between G. hirsutum and G. barbadense, the purple line indicates the collinearity of GAPDH genes between G. hirsutum and G. arboreum, the green line indicates the collinearity of GAPDH genes between G. hirsutum and G. raimondii, and the red line indicates the collinearity of the GAPDH genes of G. hirsutum.

Evolutionary tree, gene structure and motif analysis of the GAPDH genes in upland cotton

Evolutionary tree, gene structure and motif analyses based on the full-length sequences, CDSs and protein sequences of the upland cotton GAPDH genes were carried out (Fig. 4). The upland cotton GAPDH members were divided into three subgroups based on the results of evolutionary tree analysis. This finding suggests that the main difference between Groups 1 and 2 may be due to the difference in intron length between them, while the difference in Group 1 relative to Groups 2 and 3 is mainly due to the different positions of the motif. Most members of the same subgroup showed similar motifs, lengths and structures, indicating functional similarity within each group. In Group 1, the amino acid sequences of the four upland cotton GAPB members (Gh_GAPDH2, Gh_GAPDH6, Gh_GAPDH16 and Gh_GAPDH19) were longer than those of the four GAPA members (Gh_GAPDH9, Gh_GAPDH11, Gh_GAPDH21 and Gh_GAPDH23). In Group 2, Gh_GAPDH1 contained 2 motifs 7, and Gh_GAPDH26 lacked motif 4 and had no UTR. Each subgroup of protein sequences was highly conserved, but there were significant differences between the groups, especially between the sequences of Group 1 and those of Groups 2 and 3.

Figure 4 Evolutionary tree, gene structure and conserved motif analysis of the GAPDH gene family in upland cotton.

Analysis of cis-acting elements in the promoters of upland cotton GAPDH genes

Transcription factors (TFs) can regulate plant growth and development and responses to adverse stresses by regulating gene expression (Inoue & Horimoto, 2017). To further investigate the possible transcriptional regulatory mechanisms of the upland cotton GAPDH genes, the cis-acting elements in the 2,000 bp promoter sequences upstream of the 27 Gh_GAPDH genes were analyzed. The results showed that these cis-acting elements were mainly divided into hormone and stress response elements (Fig. 5). Most of the Gh_GAPDH gene promoters contained abscisic acid, methyl jasmonate (MeJA), gibberellin, low temperature, growth hormone and v-myb avian myeloblastosis viral oncogene homolog (MYB) response elements. Each upland cotton GAPDH gene promoter contained different numbers and types of cis-acting elements. The largest number of cis-acting elements in the Gh_GAPDH gene were related to ABA and MeJA, which suggests that the upland cotton GAPDH family may be mainly dependent on ABA and MeJA.

Figure 5 Analysis of cis-acting elements in the promoter region of upland cotton GAPDH family genes.

Different tissue-specific expression patterns of GAPDH genes in upland cotton

The expression levels of different GAPDH genes in different tissues of upland cotton (calycle, leaf, petal, pistil, root, stamen, stem and torus) were analyzed (Fig. 6). The results showed that Gh_GAPDH2, 6, 9, 11, 19, 21, 23 and 24 were highly expressed in the calycle; Gh_GAPDH7, 8 and 18 were highly expressed in the petal; Gh_GAPDH13, 20 and 25 were highly expressed in the root; Gh_GAPDH10, 15, 22 and 27 were highly expressed in the stamen; Gh_GAPDH4 and 17 were highly expressed in the stem; Gh_GAPDH1, 4 and 17 were highly expressed in the leaf; and Gh_GAPDH2, 9, 16 and 21 were highly expressed in the torus. The above results suggest that the gene expression patterns of members of the Gh_GAPDH gene family are not only specific but are also related to complex functions.

Figure 6 Analysis of GAPDH gene expression in different tissues of upland cotton.

Roles of the upland cotton GAPDH genes in gene expression under different abiotic stresses

Previous studies have shown that GAPDH can play a role in the response to abiotic stresses through nonglycolytic functions (Munoz et al., 2009, 2010; Guo et al., 2012; Zhao et al., 2023). Thus, it is important to investigate the expression pattern of the GAPDH genes under different stresses. We therefore analyzed the expression patterns of the GAPDH family genes after exposure to cold, heat, salt and drought stresses (Fig. 7). The expression levels of 11 genes (Gh_GAPDH2, 6, 7, 9, 11, 12, 16, 18, 19, 21 and 23) were higher than those of the other genes after cold treatment, so these 11 GhGAPDH genes may be more responsive to cold stress (Fig. 7A). After heat treatment, 10 genes (Gh_GAPDH6, 7, 9, 11, 12, 16, 18, 19, 21 and 23) tended to be expressed at high levels, but the expression levels of these 10 Gh_GAPDH genes gradually decreased as the heat treatment time increased (Fig. 7B). After salt stress treatment, nine genes (Gh_GAPDH1, 3, 4, 5, 14, 15, 17, 26 and 27) were expressed at low levels on the first branch (Fig. 7C-1), and 11 genes (Gh_GAPDH2, 6, 7, 9, 11, 12, 16, 18, 19, 21 and 23) were expressed at high levels on the third branch (Fig. 7C-3). These 11 highly expressed Gh_GAPDH genes showed a response to salt invasion at 1 h, when their expression levels reached a maximum, but their expression levels began to decline within 6 h as the duration of salt treatment increased (Fig. 7C). However, after drought stress (Fig. 7D), changes in the expression of all but six genes (Gh_GAPDH8, 15, 16, 17, 18 and 26) were observed. The expression of 11 genes (Gh_GAPDH1, 2, 4, 9, 10, 11, 19, 21, 22, 25 and 27) was upregulated and reached a maximum in the late stage of stress, indicating that these 11 upland cotton GAPDH genes are induced by drought stress and may accordingly play a role in the response of upland cotton to drought stress.

Figure 7 Analysis of GAPDH gene expression in upland cotton under different stresses.

(A) Expression pattern under cold stress. (B) Expression pattern under heat stress. (C) Expression pattern under salt stress. (D) Expression pattern under drought stress.

In brief, four genes (Gh_GAPDH9, Gh_GAPDH11, Gh_GAPDH19 and Gh_GAPDH21) were highly expressed simultaneously under cold, heat, salt and drought-induced abiotic stresses, indicating significant variability in gene expression following different stress treatments. Among these genes, the expression levels of the Gh_GAPDH11, Gh_GAPDH19 and Gh_GAPDH21 genes reached their maxima in the late stage of drought stress, while the Gh_GAPDH9 gene was upregulated from 6 h of drought stress onward, and its maximum expression occurred at 12 h. This result suggests that the Gh_GAPDH9 gene may play an important role in the early defense response when upland cotton is subjected to drought stress. Based on the above analysis, we predicted that the Gh_GAPDH9 gene might be involved in drought resistance in cotton.

Expression analysis of the Gh_GAPDH9 gene under drought stress

To further analyze the expression pattern of the Gh_GAPDH9 gene, two drought-resistant materials (Acala1517-08 and Tashigan7 hao) and two drought-sensitive materials (Xinluzhong36 and Xinluzao26) were selected and subjected to qRT‒PCR analysis. We examined the relative expression levels of the Gh_GAPDH9 gene at seven time intervals after 15% PEG6000 treatment (Fig. 8A). The results showed that the expression levels of drought-resistant and drought-sensitive materials were significantly different at 0.5, 1 and 3 h. The expression levels in the drought-resistant materials Acala1517-08 and Tashigan7 hao showed a continuous increase from 0 to 3 h and decreased after 6 h of stress. The expression of this gene was downregulated in the drought-sensitive materials Xinluzhong36 and Xinluzao26 from 0.5 to 24 h, and it was barely expressed at the beginning of the drought treatment.

Figure 8 qRT‒PCR analysis of the Gh_GAPDH9 gene under drought stress.

(A) Analysis of the expression levels of Gh_GAPDH9 in drought-resistant and drought-sensitive cotton materials. The associated significance levels (p values) are provided above each histogram. Asterisks (* and **) represent significant effects at the p < 0.01 and p < 0.001 levels, respectively. (B) Analysis of the tissue-specific expression levels of Gh_GAPDH9.

The Gh_GAPDH9 gene was expressed at different levels in different tissues, and understanding its expression in different tissues provides a basis for better elucidating the molecular mechanism of the gene. The expression of the Gh_GAPDH9 gene was greatest in the drought-tolerant material at 3 h of drought stress, and it is hypothesized that cotton may respond to 15% PEG6000 stress after 3 h of exposure. Therefore, we also examined the involvement of the Gh_GAPDH9 gene in the response to drought in other tissues of cotton (Fig. 8B). The expression levels of Gh_GAPDH9 were highly significantly different only in leaves (p < 0.01), showing highly significant increases in the leaves of both drought-resistant materials (Acala1517-08 and Tashigan7 hao) and drought-sensitive materials (Xinluzhong36 and Xinluzao26) after PEG drought stress.

Functional analysis of the Gh_GAPDH9 gene in response to drought stress

In this experiment, we constructed a pTRV2-GhGAPDH9 VIGS vector. At 10 d after cotton cotyledon injection, pTRV2-CLA plants showed bleaching, confirming the success of the silencing system in this assay (Fig. 9A). The expression of Gh_GAPDH9 was significantly lower in the silenced plants than in the control plants, indicating that the Gh_GAPDH9 gene was effectively silenced (Fig. 9B). The control and silenced plants were then divided into two groups, one subjected to normal irrigation treatment (Control) and one subjected to natural drought treatment (Drought). The plants were photographed and their characteristics were recorded before drought (0 d) and during drought treatment (15 d). The results showed that there was no significant difference between the silenced plants and the control plants before stress treatment, and after stress treatment, the GhGAPDH9-silenced plants showed more severe leaf wilting than the control plants (Fig. 9C).

Figure 9 Drought resistance of Gh_GAPDH9 gene.

(A) pTRV2-CLA albino plants. (B) Detection of Gh_GAPDH9 gene silencing efficiency. Asterisks (* and ***) represent significant effects at the p < 0.05, and p < 0.001 levels, respectively. (C) Drought phenotypes of control cotton plants (pTRV2) and silenced cotton plants (pTRV2-GhGAPDH9). (D) Changes in physiological indicators after drought stress in control plants (pTRV2) and silenced plants (pTRV2-GhGAPDH9). (E) Detection of the Gh_GAPDH9 gene expression efficiency. (F) Drought phenotype of wild-type (WT) and overexpressed Arabidopsis (pCAMBIA3301-GhGAPDH9).

To analyze the physiological mechanism of Gh_GAPDH9 gene silencing in response to drought stress, we examined the MDA, Pro, chlorophyll, Chla and Chlb contents of silenced GhGAPDH9 and control plants before and after drought treatment. The results showed that the contents of these five indicators did not differ significantly in silenced Gh_GAPDH9 and control plants before stress treatment, whereas after stress treatment, silenced Gh_GAPDH9 plants showed significantly higher MDA content and significantly lower Pro, chlorophyll and Chlb contents compared with the control plants (Fig. 9D). To further determine the function of Gh_GAPDH9, we transferred it to A. thaliana. The expression of Gh_GAPDH9 in transgenic strains was also confirmed by qRT‒PCR (Fig. 9E). Compared with wild-type (WT), the leaf wilting and dryness of transgenic strains under drought stress were lower, indicating that Gh_GAPDH9 is a positive regulator of drought resistance (Fig. 9F).

Discussion

The GAPDH gene family plays an important role in plant growth and development and resistance to adverse stress conditions. In recent years, the identification and functional studies of plant GAPDH family genes have attracted increasing attention. With the continuous development of sequencing technologies, the genome of cotton has been refined and updated (Wang et al., 2012a, 2019; Hu et al., 2019), and tandem repeat events and large-block repeat events are thought to be the main reasons for the expansion of gene families in the genome (Rui et al., 2022; Wang et al., 2022). To date, GAPDH family genes have been well studied in Arabidopsis (Vescovi et al., 2013), wheat (Zeng et al., 2016) and soybean (Zhao et al., 2023) crops, whereas no systematic analysis of the GAPDH gene family in cotton has been reported. To better understand the functions of this family, we comprehensively identified and analyzed the Gh_GAPDH genes. The results of the present study showed no tandem or segmental duplication events in the GAPDH gene family (Fig. 1). Fourteen, 14, 27 and 29 GAPDH genes were identified in four cotton varieties (G. arboreum, G. raimondii, G. hirsutum and G. barbadense), in accordance with a process of chromosome doubling and gene evolution leading to differences in the number of GAPDH genes in cotton (Deng, Chen & Qu, 2022). Based on the evolutionary relationships of the GAPDH family members of Arabidopsis (Marri et al., 2005; Anoman et al., 2015; Mirko et al., 2013), we classified the GAPDH genes of cotton into three categories according to isoform classification: GAPA/B, GAPC and GAPCp (Fig. 2).

There are four domesticated cultivars in the genus. Gossypium herbaceum (A1 genome) and G. arboreum (A2 genome) are diploid A genomes, and G. hirsutum (AD1 genome) and G. barbadense (AD2 genome) are heterotetraploid AD genomes. Compared with diploid cultivated cotton, tetraploid cotton has obvious advantages in fiber length and quality. The heterotetraploid AD genome is formed by natural hybridization and chromosome doubling of the D and A genomes. Scholars recognize that its D subgenome donor is Raymond’s cotton (D5 genome), and the A subgenome donor is the G. arboreum A2 genome. Through chromosomal localization collinear analysis, we found that the GAPDH genes were not evenly distributed across the chromosomes in cotton, in which no GAPDH genes were distributed on chromosomes At/Dt-chr06, At/Dt-chr08 and At/Dt-chr12; a total of 759 covariate pairs were identified. In addition, the GAPDH family gene sequence of the terrestrial cotton A subgenome is collinear with the sequence of G. arboreum and G. barbadense, and the GAPDH gene sequence in the D subgenome is collinear with one sequence in G. raimondii and G. barbadense (Fig. 3). These results suggest that terrestrial cotton GAPDH genes are relatively conserved during the evolution of cotton (Fig. 3). The large number of gene family members and similar conserved motifs ensure that the genes are functionally linked to each other, and changes in the structures and sequences of differentially expressed genes can lead to functional divergence (Deng, Chen & Qu, 2022). An analysis of Gh_GAPDH gene structure and motifs revealed that differences in intron length and the locations of motifs may have contributed to the differences in the three subgroups (Fig. 4). The members of a given subgroup present similar functions, and Brinkmann et al. (1989) showed that the photosynthetic GAPDH categories are GAPA and GAPB, which exhibit high (80%) sequence homology, although the B subunit is three dozen amino acids longer at the C-terminus than the A subunit. According to our results, the amino acid sequences of the four GhGAPB genes are indeed longer than those of the four GhGAPA genes, similar to previous findings (Zeng et al., 2016; Vescovi et al., 2013); therefore, it is speculated that cotton may show a similar evolutionary trajectory to Arabidopsis and wheat, for example.

The type and number of cis-acting elements within the promoter regions of genes can laterally indicate their related functions (Tong et al., 2020), and many hormones associated with MeJA and ABA were identified via an analysis of the promoter regions of Gh_GAPDH family genes. ABA is the most important hormone in plants facing abiotic stresses (Ruan et al., 2019), further indicating that these genes may exert their effects through hormonal regulation.

GAPDH genes not only play a role in plant growth and development but may also be involved in the regulation of abiotic stresses in plants. We analyzed the RNA-seq data for the Gh_GAPDH genes in eight tissues and found that more Gh_GAPDH genes were highly expressed in calyx, petal, root and stamen tissues. For example, Gh_GAPDH13, 20 and 25 were expressed at high levels in the root system, suggesting that they may be involved in root development, similar to the finding that Arabidopsis GAPCp gene deficiency leads to altered root development (Munoz et al., 2009). Abiotic stress is a major cause of reduced crop quality and yields (Du et al., 2022). We analyzed the expression patterns of GAPDH family genes after cold, heat, salt and drought stresses and found that four genes (Gh_GAPDH9, 11, 19 and 21) were highly expressed under abiotic stresses, indicating that these four GAPDH genes are likely associated with the response to abiotic stresses under different stress treatments. The Gh_GAPDH9 expression trend was completely opposite under cold and high-temperature stress conditions, and the expression was downregulated under salt stress. Only the GhGAPDH9 gene was found to play an important role in early defense responses. Gh_GAPDH11 expression was downregulated under salt stress and increased under other stressful conditions. Gh_GAPDH19 expression was downregulated under high temperature and salt stress conditions and increased under cold and drought stress conditions. These results suggest that the same gene may play different roles under different stress conditions. In addition, Yang found that the GmGAPDH protein plays a certain regulatory role in the process of soybean mosaic virus infection in soybean (Yang, 2023), and that the StGAPDH gene is closely related to maize Setosphaeria turcica infestation and disease development during its pathogenicity in maize (Zhang et al., 2023). This finding also provides important information for our future research on the function and mechanism of Gh_GAPDH under biological stress conditions in cotton.

To further determine whether Gh_GAPDH9 gene expression levels were associated with drought stress, expression pattern analysis was performed on drought-resistant and drought-sensitive materials in this study. The results showed that Gh_GAPDH9 was highly expressed in early drought-tolerant materials, while its expression was largely absent in drought-sensitive materials (Fig. 8A). Previous studies have shown that antioxidant regulation and osmoregulation are essential for drought adaptation in plants (Kaur & Asthir, 2017). Under drought conditions, ROS levels can increase rapidly (Miller et al., 2010), but excess ROS can lead to impaired membrane integrity and enzyme inhibition (Impa, Nadaradjan & Jagadish, 2012) and increased MDA contents (Pandey et al., 2010), and an excessive MDA content can in turn cause damage to plants (Liu et al., 2021). Pro accumulation protects cells from damage through the action of Pro as an osmotic agent and free radical scavenger (Kavi & Sreenivasulu, 2014), and changes in its content can directly reflect the plant’s resistance to stress. We found that after silencing Gh_GAPDH9, the silenced plants showed more severe leaf wilting (Fig. 9C), a highly significant increase in MDA content and significant decreases in Pro, chlorophyll and Chlb contents relative to the control plants after stress treatment (Fig. 9D). These results suggest that Gh_GAPDH9 is a positively regulated gene that contributes to drought tolerance by affecting antioxidant capacity and osmoregulation in cotton. Overexpression of the Gh_GAPDH9 gene resulted in lower leaf wilt in the transgenic strain (Fig. 9F), further indicating that Gh_GAPDH9 (GhGAPA2) is a positive regulator of drought tolerance. Shiraku et al. (2022) found that the interaction of the GhVDAC1, GhGAPA and GhLEA3 genes contributed to enhanced drought and salt tolerance in cotton, similar to our results.

Conclusion

In this study, we identified a total of 84 GAPDH genes in four cotton varieties, 27 of which were present in upland cotton. Evolutionary tree and covariance analyses showed that the cotton GAPDH family could be divided into three categories, GAPA/B, GAPC and GAPCp, with relatively conserved evolutionary processes. Expression analysis revealed that four genes (Gh_GAPDH9, 11, 19 and 21) were highly expressed under cold, heat, salt and drought induction. qRT‒PCR revealed that the Gh_GAPDH9 gene was highly expressed in drought-resistant material and leaf tissue. In addition, silencing of the Gh_GAPDH9 gene under drought stress resulted in increased MDA content in cotton leaves, and overexpression of Gh_GAPDH9 in transgenic Arabidopsis led to lower leaf wilt, indicating that Gh_GAPDH9 can positively regulate drought tolerance in cotton plants. In conclusion, Gh_GAPDH9 could be a candidate gene for drought tolerance breeding. The present study not only provides the first insight into the GAPDH gene family in cotton but also lays a foundation for future in-depth analysis of the molecular mechanisms of drought tolerance in cotton.

Supplemental Information

Supplemental Information 1 Information on the GAPDH gene family in G. hirsutum.

Click here for additional data file.

Supplemental Information 2 qRT-PCR raw data.

Click here for additional data file.

Additional Information and Declarations

Competing Interests

Author Contributions

Data Availability

The authors declare that they have no competing interests.

Shiwei Geng conceived and designed the experiments, performed the experiments, analyzed the data, prepared figures and/or tables, authored or reviewed drafts of the article, and approved the final draft.

Shengmei Li conceived and designed the experiments, performed the experiments, analyzed the data, prepared figures and/or tables, authored or reviewed drafts of the article, and approved the final draft.

Jieyin Zhao performed the experiments, prepared figures and/or tables, and approved the final draft.

Wenju Gao performed the experiments, prepared figures and/or tables, and approved the final draft.

Qin Chen performed the experiments, prepared figures and/or tables, and approved the final draft.

Kai Zheng performed the experiments, analyzed the data, prepared figures and/or tables, and approved the final draft.

Yuxiang Wang analyzed the data, prepared figures and/or tables, and approved the final draft.

Yang Jiao analyzed the data, authored or reviewed drafts of the article, and approved the final draft.

Yilei Long conceived and designed the experiments, authored or reviewed drafts of the article, and approved the final draft.

Pengfei Liu conceived and designed the experiments, authored or reviewed drafts of the article, and approved the final draft.

Yanying Qu conceived and designed the experiments, authored or reviewed drafts of the article, and approved the final draft.

Quanjia Chen conceived and designed the experiments, authored or reviewed drafts of the article, and approved the final draft.

The following information was supplied regarding data availability:

The raw data are available in the Supplemental File.

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
