# Peer review of "Glyceraldehyde-3-phosphate dehydrogenase Gh_GAPDH9 is associated with drought resistance in Gossypium hirsutum"

_PeerJ, doi:10.7717/peerj.16445_

## Round 0.1 · original submission · Major Revisions

Dear Authors

According to the reviewer's comments, this manuscript cannot be accepted for publication yet. It needs a major revision to be reconsidered for publication. The authors are invited to revise the paper considering all the suggestions made by the reviewers. Please note that requested changes are required for publication.
In addition, there are concerns about the manuscript's grammar, usage, and overall readability. Therefore, request that you revise the text to fix the grammatical errors and improve the overall readability of the text.
With Thanks

Reviewer 1 ·

Basic reporting

Overall, the article is well-written and provides a good background on the importance of cotton as a cash crop and the detrimental effects of drought stress on cotton growth and yield. It also introduces the role of glyceraldehyde-3-phosphate dehydrogenase (GAPDH) as a key enzyme in glycolysis and its emerging functions in non-glycolytic processes related to stress response and growth and development pathways.

However, please consider the following points:

1. The introduction could benefit from clearer organization and flow of information. It starts with the general context of cotton as a cash crop and the impact of drought stress, but then jumps directly into discussing GAPDH and its functions in various plant species. It would be helpful to provide a smoother transition and clearly state the objective of the study earlier in the introduction.
2. The introduction includes several references to support the statements made about the functions of GAPDH in different plant species. It would be beneficial to include more recent references to reflect the latest research in the field. Additionally, it would be helpful to provide more specific details about these studies, such as the methods used and key findings, to give readers a better understanding of the research landscape.
3. The introduction briefly mentions that the GAPDH genes of cotton have not been identified, but it does not explicitly highlight the gap in knowledge that the study aims to address. Clearly stating the research gap and the specific objectives of the study would provide a stronger rationale for the research and create a stronger context for the subsequent sections.
4. The introduction briefly mentions that genome-wide identification of the cotton GAPDH gene family was performed and that bioinformatics analysis was used to explore various aspects of the gene family. However, it would be beneficial to provide more details about the specific methods and tools used for genome-wide identification, phylogenetic analysis, gene structure analysis, and expression profiling. This would help readers understand the technical approach of the study.
5. The introduction could conclude with a clearer statement on the significance and potential implications of the study. What are the potential applications or benefits of understanding the functions of GAPDH genes in cotton, especially in relation to drought stress? This would help readers understand the broader importance of the research.

Experimental design

Plant material:
- Specify each cotton material's specific cultivar or accession numbers.
- Provide a brief justification for the selection of drought-resistant and drought-sensitive cotton materials.

Genomic and proteomic data analysis:
- Mention the source of the genomic and proteomic data obtained for G. arboreum, G. raimondii, G. hirsutum, and G. barbadense.
- Include the rationale for using the cryptomarkov model and HMMER software for analysis.
- Provide references for the tools and databases used in the analysis.

Phylogenetic and covariance analysis:
- Describe the purpose or significance of performing phylogenetic and covariance analysis on the GAPDH gene family in cotton.
- Specify the specific version of the MEGA software used for multiple sequence alignment.

Chromosomal localization, gene structure, and motif analysis:
- Clarify the importance of analyzing chromosomal localization, gene structure, and motif analysis in the context of the study.
- Provide more details (e.g., version) on the software or tools used for chromosomal localization and motif analysis.

Analysis of cis-acting elements:
- Explain the relevance of analyzing cis-acting elements in the promoter sequences of the GAPDH gene family.
- Describe any specific criteria or considerations used for predicting the cis-acting elements.

RNA-seq analysis:
- Specify the tissues used for transcriptomic data analysis in upland cotton.
- Include the accession number or reference for the upland cotton genome used as the reference.
- Briefly explain the FPKM method and its advantages in assessing gene expression.

qRT-PCR and VIGS/overexpression:
- Reference previously reported similar work that used the qRT-PCR procedure on cotton.
- Provide a concise overview of the purpose and significance of using VIGS and overexpression techniques.
- Clarify the specific traits or physiological indicators measured (MDA, Pro, Chl) and their relevance to drought response.

Validity of the findings

While the results of the study provide valuable insights into the role of the Gh_GAPDH9 gene in cotton's response to drought stress, there are some aspects that can be criticized:
- Limited sample size: It would be more robust to include a larger sample size and replicate the experiments to ensure the reliability and generalizability of the findings.

- Lack of functional validation: The study primarily relied on gene expression analysis and gene silencing techniques to infer the role of the Gh_GAPDH9 gene in cotton. However, more comprehensive functional validation, such as overexpression of the gene and complementation studies, would strengthen the conclusions and provide more direct evidence of its role.

- Incomplete stress conditions: The study focused on analyzing the expression patterns of GAPDH genes under cold, heat, salt, and drought stresses. While these are important abiotic stresses, there are other relevant stress conditions that could be considered, such as pathogen infection or nutrient deficiency, to provide a more comprehensive understanding of the gene's response to stress, at least to be covered in the discussion part by proper litreature.

- Limited exploration of downstream effects: The study mainly investigated the physiological parameters and antioxidant enzyme activities in GhGAPDH9-silenced plants. Further analysis of downstream effects, such as gene expression changes in pathways related to stress tolerance or additional physiological parameters, would provide a more holistic view of the gene's impact on cotton's response to drought stress, at least from the RNAseq perspective.

- Limited discussion of evolutionary implications: While the study briefly touched upon the evolutionary relationships of GAPDH genes in cotton, further exploration of the evolutionary implications of the findings would be beneficial. Understanding the evolutionary history and conservation of GAPDH genes across different plant species could enhance our understanding of their functional significance.

Reviewer 2 ·

Basic reporting

The submitted manuscript determined the cotton glyceraldehyde-3-phosphate dehydrogenase (GAPDH), an enzyme involved in glycolysis, using genome-wide analysis. The selected GAPDH, GhGAPDH9, was characterised by its functional role in response to drought stress. The generated data might help understand the role of GAPDH in drought tolerance. Although the manuscript deals with interesting and important topics, some improvements are desirable.

English
Moderate linguistic (English) corrections or polishing are desirable. In addition, the submitted version has numerous sentence framing/ phrase construction issues.

Introduction
L45-53: These few sentences could be reorganised for clarity. For example, L46 highlights that drought could affect cotton yield. However, L50 and L52 seem to repeat the same information.
L58: “is expressed at high levels in almost all tissues”. Is this referring to normal or stressful conditions?
L59: “However, recent studies have shown that….”. This sentence did not explain or support the previous statement. So, why should the authors use “However, recent studies…”? Perhaps, the authors could discuss previous studies related to GAPDH to show the importance of this enzyme in glycolysis before discussing its role in non-glycolysis.
L63-70: The authors could select 1-2 of these examples to elaborate further or explain how GAPDH could enhance the stress tolerance in these crops instead of listing them.
L65: “is upregulated” change to “was upregulated”. Similar to L66.
L67-68: What is PLD?
L72: “is closely related” may not accurately convey the intended meaning. Perhaps, this sentence could be rephrased to “the members of the GAPDH gene family have been demonstrated to be associated with drought and salt stress”.
L76: “…the analysis of additional GAPDH gene families is key to resolving the functions of this gene family”. How could additional GAPDH genes resolve their function?
L80: No further discussion on non-phosphorylated types?

Experimental design

Overall, the work appears to have been performed in an acceptable manner, although some experiments required further clarification.
L97: “Based on previous research”. Which studies? Please provide references.
L97-98: Do they have the voucher number?
L107: “Water was changed” or Hoagland’s nutrient solution? Also, how frequent is “change regularly”?
L110: How do the authors know 15% PEG could stimulate drought stress conditions?
L113: Why was Arabidopsis used as WT?
L153: What are the criteria for filtering those data?
L162: Since L96 already has “Plant material”, the word “plant material” probably can be removed.
L163: “was the same as” change to “was performed according to Li et al. (2022)”.
L169: Please provide the qRT-PCR running conditions and information about sample replicates.
L180: “MDA, Pro and Chl”. Please provide the full name for the first time mentioned.
L184: “overexpressing Arabidopsis” change to “GhGAPDH9-overexpressing Arabidopsis”.
L186: “natural drought”. Please specify how this was conducted.
Please provide a section on statistical analysis.

Validity of the findings

Results
L191-195: The first and second sentences seem not properly gelled. While the first sentence highlights the changes in the copy number in the GAPDH gene family were determined. However, the second sentence provides information about protein domains. Would the information in L195-197 is more suitable for placing after the first sentence? Perhaps, the authors could highlight what has been done to achieve the main goal step-by-step.
L205: Is there Table 1? I’m afraid I did not see/receive this.
L210: “Previous studies….”. Please provide references to support such claims.
L221: “four different cotton species”. Please specify.
L258-262: Please shorten or separate this sentence into two.
L263: Please provide the full name for the first mentioned.
L280: “Previous studies….”. Please provide references.

Discussion
In general, the Discussion section requires improvements. The current version often repeats what has been discussed/mentioned in the Results section.
L353-359: Suggest the authors simplify or shorten this information, as it has been mentioned in the introduction.

Figure 2 caption: The caption seems too simplified without much information. What shapes represent which groups?

---

## Round 0.2 · Minor Revisions

Dear Authors

The manuscript still needs a minor revision to be reconsidered for publication. The authors are invited to revise the paper considering all the suggestions made by the reviewer. Please note that requested changes are required for publication.

Furthermore, the manuscript exhibits issues with the English language, which hinders the reader's comprehension. I recommend that the authors seek assistance from a fluent English speaker or professional language editor to enhance the overall clarity and readability of the manuscript. A well-written paper is essential for effective communication in the academic community.
With Thanks

**Language Note:** The Academic Editor has identified that the English language must be improved. PeerJ can provide language editing services - please contact us at [email protected] for pricing (be sure to provide your manuscript number and title). Alternatively, you should make your own arrangements to improve the language quality and provide details in your response letter. – PeerJ Staff

Reviewer 2 ·

Basic reporting

The authors have improved the manuscript and addressed most of my comments. However, some minor improvements might be desirable.

Although the authors have amended the manuscript, some language polishing is still required. For example,
L10: "adverse stresses conditions" change to "adverse stress conditions".
L309: "...and the motifs of Groups 1 and 2 did not differ". This can be in another sentence.
L327: Small capital for "Jasmonate".
L355: "...on the first branch". What does the first branch mean? The first column or the first hour? Similar to L356.

Experimental design

L201-204: "Of course, different samples may be sequenced at different depths, and the deeper the depth, the more corresponding reads there are dividing the results by the number of respective libraries (measured in one million reads) provides a good measure of the relative expression of the same gene in two different samples". This sentence seems a bit lengthy and difficult to understand. Perhaps the authors could revise for clarity.
L210: Delete "as previously reported".
L232: Since "MDA, Pro and Chl" were first mentioned here, their full name should be inserted. The subsequent repeats could use abbreviations.
L235: "...by referring to the method of Fan (Fan et al., 2021)" change to "according to Fan et al. (2021)".

Validity of the findings

The revised version is statistically sound. Information about replicates has been provided.

---

## Round 0.3 · Minor Revisions

Dear Authors

The manuscript needs a minor revision to be reconsidered for publication. Authors need to consider this citation; furthermore, they need to explain how their study is different and cite if appropriate:

Sun J, Cui H, Wu B, Wang W, Yang Q, Zhang Y, Yang S, Zhao Y, Xu D, Liu G, Qin T. Genome-Wide Identification of Cotton (Gossypium spp.) Glycerol-3-Phosphate Dehydrogenase (GPDH) Family Members and the Role of GhGPDH5 in Response to Drought Stress. Plants (Basel). 2022 Feb 22;11(5):592. doi: 10.3390/plants11050592."

Please note that requested changes are required for publication.
Best Regards

Reviewer 2 ·

Basic reporting

The authors have revised and improved the manuscript accordingly. I have no further comments on the revised version.

Experimental design

No comment.

Validity of the findings

No comment.

Additional comments

No comment.

---

## Round 0.4 · accepted · Accept

Dear Authors

I am pleased to inform you that after the last round of revision, the manuscript has been improved a lot, and it can be accepted for publication.

Congratulations on the acceptance of your manuscript, and thank you for your
interest in submitting your work to PeerJ

With Thanks